# Replication fork binding triggers structural changes in the PriA helicase that govern DNA replication restart in *E. coli*

Alexander T. Duckworth [1], Peter L. Ducos [2,3,5], Sarah D. McMillan [1,5], Kenneth A. Satyshur [1], Katelien H. Blumenthal[1], Haley R. Deorio [1], Joseph A. Larson [1], Steven J. Sandler [4] ✉, Timothy Grant [2,3] ✉ & James L. Keck [1] ✉

Bacterial replisomes often dissociate from replication forks before chromosomal replication is complete. To avoid the lethal consequences of such situations, bacteria have evolved replication restart pathways that reload replisomes onto prematurely terminated replication forks. To understand how the primary replication restart pathway in *E. coli* (PriA-PriB) selectively acts on replication forks, we determined the cryogenic-electron microscopy structure of a PriA/PriB/replication fork complex. Replication fork specificity arises from extensive PriA interactions with each arm of the branched DNA. These interactions reshape the PriA protein to create a pore encircling single-stranded lagging-strand DNA while also exposing a surface of PriA onto which PriB docks. Together with supporting biochemical and genetic studies, the structure reveals a switch-like mechanism for replication restart initiation in which restructuring of PriA directly couples replication fork recognition to PriA/PriB complex formation to ensure robust and high-fidelity replication re-initiation.

Replication of the circular genomes found in most bacteria starts with sequence-dependent loading of two DNA replication complexes (replisomes) onto a single origin of replication. The replisomes proceed in opposite directions around the chromosome until they reach a programmed termination sequence, producing two identical genome copies[1]. However, replisomal progress is frequently challenged by encounters with barriers such as damaged DNA templates or proteins that are tightly bound to the chromosome. Collisions with these obstacles can stall replisomes and, at least once per cell cycle, cause the replisome to dissociate from its DNA replication fork[2–4]. Premature replication termination leaves an incompletely replicated chromosome, which is lethal to the cell if left unrepaired.

Replication failure can happen anywhere along the chromosome, which precludes the use of sequence-specific repair mechanisms to rescue prematurely terminated replication reactions. Instead, bacteria have evolved specialized DNA replication restart pathways that reload replisomes onto replication forks in a structure-specific manner[5–8]. Replication restart processes recognize replication forks within a vast excess of genomic DNA, remodel the forks to prepare them for replicative helicase (DnaB in *E. coli*) reloading, and load DnaB onto the single-stranded (ss) lagging strand. Replisome reassembly occurs spontaneously once DnaB is reloaded on the replication fork[9–12]. As with canonical replication, the fidelity of replication restart is of great importance since dysregulated replication initiation leads to genomic instability[13,14]. How replication restart processes act with the specificity needed for selective activity on replication forks is poorly understood.

In *E. coli*, replication restart is facilitated by several proteins organized into multiple pathways[6,7]. PriA, which is the only

[1]Department of Biomolecular Chemistry, University of Wisconsin-Madison, Madison, WI 53706, USA. [2]Department of Biochemistry, University of Wisconsin-Madison, Madison, WI 53706, USA. [3]John and Jeanne Rowe Center for Research in Virology, Morgridge Institute for Research, Madison, WI 53715, USA. [4]Department of Microbiology, University of Massachusetts at Amherst, Amherst, MA 01003, USA. [5]These authors contributed equally: Peter L. Ducos, Sarah D. McMillan. ✉e-mail: sandler@microbio.umass.edu; tim.grant@wisc.edu; jlkeck@wisc.edu

replication restart factor found in all bacteria, acts as a "first responder" protein, recognizing abandoned DNA replication forks in a structure-specific manner and recruiting other factors needed for replication re-initiation[8,15]. Accordingly, *priA*-null *E. coli* display genomic instability phenotypes that include slow growth in rich media, cell division defects, and induction of the genomic stress response SOS regulon[16–19]. The PriA-PriB pathway serves as the predominant replication restart pathway in *E. coli*, and is particularly important in cells with high levels of double-stranded (ds) DNA breaks[20–23]. Recombinational repair of dsDNA breaks produces D-loop DNA structures that mimic abandoned replication forks with ssDNA lagging strands; such structures are preferentially processed by the PriA-PriB pathway in vivo[20,23].

Abandoned replication fork recognition by PriA and assembly of the PriA/PriB protein complex are tightly linked steps that regulate initiation of the PriA-PriB pathway. PriA is a 3′−5′ DNA helicase that comprises six domains: a 3′ binding domain (3′BD), a winged-helix domain (WH), two helicase domains (HD1 and HD2), a cysteine-rich region (CRR), and a C-terminal domain (CTD)[24]. In mechanisms that are not well understood, PriA's domains cooperate to bind replication fork structures selectively and with high affinity, and unwind the lagging strand when it is dsDNA[25–27]. Binding of PriA to a replication fork triggers PriB docking, with PriB binding ~100-fold tighter to the PriA/replication fork complex than to PriA alone[28–31]. PriB, which is a homodimeric paralog of ssDNA-binding protein (SSB), stimulates the DNA unwinding activity of PriA and forms a stable complex with PriA bound to replication forks that have ssDNA lagging strands[28,29,32,33]. Once assembled, the PriA/PriB complex recruits DnaT to form the complex necessary to load DnaB from the DnaB/DnaC helicase/loader complex[9,28,29].

To investigate the molecular mechanisms underlying DNA replication restart initiation, we determined a 3.2 Å global resolution structure of the *E. coli* PriA/PriB/replication fork complex using cryogenic-electron microscopy (cryo-EM) single-particle analysis. PriA extensively interacts with each arm of the forked DNA within the structure, conferring the specificity needed for selective binding to DNA replication forks and D-loops. Replication fork binding induces a dramatic conformational change in PriA in which the CRR domain pivots ~85° and the HD2 domain rotates ~15° to create a pore enclosing lagging strand ssDNA. This change also exposes a surface on the PriA-CRR domain onto which PriB docks, positioning PriB to bind ssDNA extending from the lagging strand pore. The structure thus elucidates a switch-like mechanism for replication restart initiation in which replication fork binding remodels the structure of PriA to trigger PriB recruitment and replication re-initiation on replication fork structures. Biochemical and genetic experiments examining the PriA/PriB interface further support this concerted mechanism for efficient and high-fidelity PriA-mediated replication restart.

## Results

### Cryo-EM structure of the *E. coli* PriA/PriB/replication fork complex

DNA replication restart in the PriA-PriB pathway is initiated through structure-specific replication fork binding by PriA and assembly of the PriA/PriB protein complex[27–29]. To probe the physical basis underlying this process, we used cryo-EM single-particle analysis to determine the structure of an *E. coli* PriA/PriB/replication fork complex. The DNA replication fork included a 25-base pair parental arm, a 15-base pair leading strand, and a 15-nucleotide ssDNA lagging strand (Supplementary Table 1, Fork 5). The nascent leading strand anneals near the fork junction to facilitate interaction with the PriA-3′BD while the lagging strand was maintained as ssDNA to promote PriB association with the PriA/DNA complex[32,34]. Cryo-EM single-particle analysis revealed one major class of the PriA/PriB/replication fork complex along with two additional minor classes (Supplementary Fig. 1A).

The major PriA/PriB/replication fork cryo-EM class represented ~80% of observed particles. Analysis of this class resulted in a 3.2 Å global resolution map that included EM density for a PriA monomer, a PriB dimer, and each arm of the replication fork DNA substrate (Fig. 1A and Supplementary Fig. 1B, D). Prior crystal structures of PriA[24,27] and PriB[35–38] were readily fit into the map, with notable differences described below. Rigid-body fitting of domains and manual refinement resulted in a well fit model with good bond geometry (Fig. 1B and Supplementary Table 2).

Inspection of the PriA/PriB/replication fork structure showed that the position of the PriA-CRR was significantly altered by complex formation, pivoting ~85° relative to its position in prior crystal structures (Fig. 1C)[24,27]. Rearrangement of the PriA-CRR and a more subtle ~15° rotation of the PriA-HD2 created an unexpected structural feature in PriA: a-25-Å long pore that is framed by the CRR, HD2, and CTD domains. Well-resolved EM density showed lagging strand ssDNA from the replication fork threaded through the pore, with 8-9 nucleotides stretching from the replication fork junction to the end of the pore. The ~15 Å width of the pore is sufficient to accommodate ssDNA but not dsDNA, which could indicate that the binding arrangement requires ssDNA on the lagging strand. PriA residues within the pore and additional residues from the 3′BD and HD1 domains combine to form an extensive, evolutionarily conserved binding surface for lagging-strand ssDNA (Supplementary Fig. 2A). In total, 42 PriA/lagging strand contacts are observed in the structure (Supplementary Fig. 2B, assessed using DNAproDB[39]).

In addition to lagging strand binding, PriA binds the leading strand and parental arms of the replication fork to create a multi-point binding surface that recognizes structural features characteristic of abandoned replication forks and D-loops (Fig. 1 and Supplementary Fig. 2). The PriA-3′BD engages the leading strand dsDNA, with PriA displacing the 3′ end of the nascent leading strand and binding it in a specialized pocket as has been observed previously[27,34]. Residues from the PriA-CTD also contribute to leading strand binding (Supplementary Fig. 2). Parental dsDNA is bound by the HD1, 3′BD, and WH domains of PriA (Supplementary Fig. 2). A loop from PriA-HD1 with noted roles in coupling DNA binding and ATPase activities[40] is wedged into the fork junction at the base of the parental duplex and the 3′BD contacts the lagging strand within the parental duplex near the fork junction. The conformation of the loop is not altered by replication fork binding. The WH domain docks into the major groove of parental dsDNA 10–20 base pairs from the replication fork junction, which correlates well with prior site-specific crosslinking experiments[27]. The PriA-WH domain and parental dsDNA are not as well resolved as the remainder of the structure (Fig. 1A and Supplementary Fig. 1B), which could be due to previously noted dynamics of the PriA-WH[27] and DNA mobility. Focused classification was required to obtain density allowing fitting of these regions. The modest resolution of the maps limited modelling to low-confidence rigid-body fitting of the PriA-WH and DNA. Together, multiple lagging strand, leading strand, and parental duplex binding elements provide the underlying physical mechanism for PriA's highly specific and stable interaction with replication forks.

In addition to opening a ssDNA-binding pore, a second notable impact of PriA-CRR pivoting is that it exposes a surface of the domain onto which PriB docks, directly linking PriA replication fork recognition with PriA/PriB protein complex formation (Fig. 1C, D). PriA/PriB interaction in this arrangement buries ~1600 Å² of surface area (calculated using the PISA server[41]) and is supported by a network of well conserved and electrostatically compatible residues from both proteins (Fig. 2 and Supplementary Fig. 1C). These include Asp438, Ser481, Thr482, and His483 from PriA interacting with His26, Glu39, Arg44, and His81 from PriB. A prior report noting that Glu39Ala and Arg44Ala *E. coli* PriB variants fail to interact with PriA[29] is consistent with the observed interface. Binding to the CRR places PriB at the edge of the PriA pore entrance where PriB directly engages ssDNA extending from

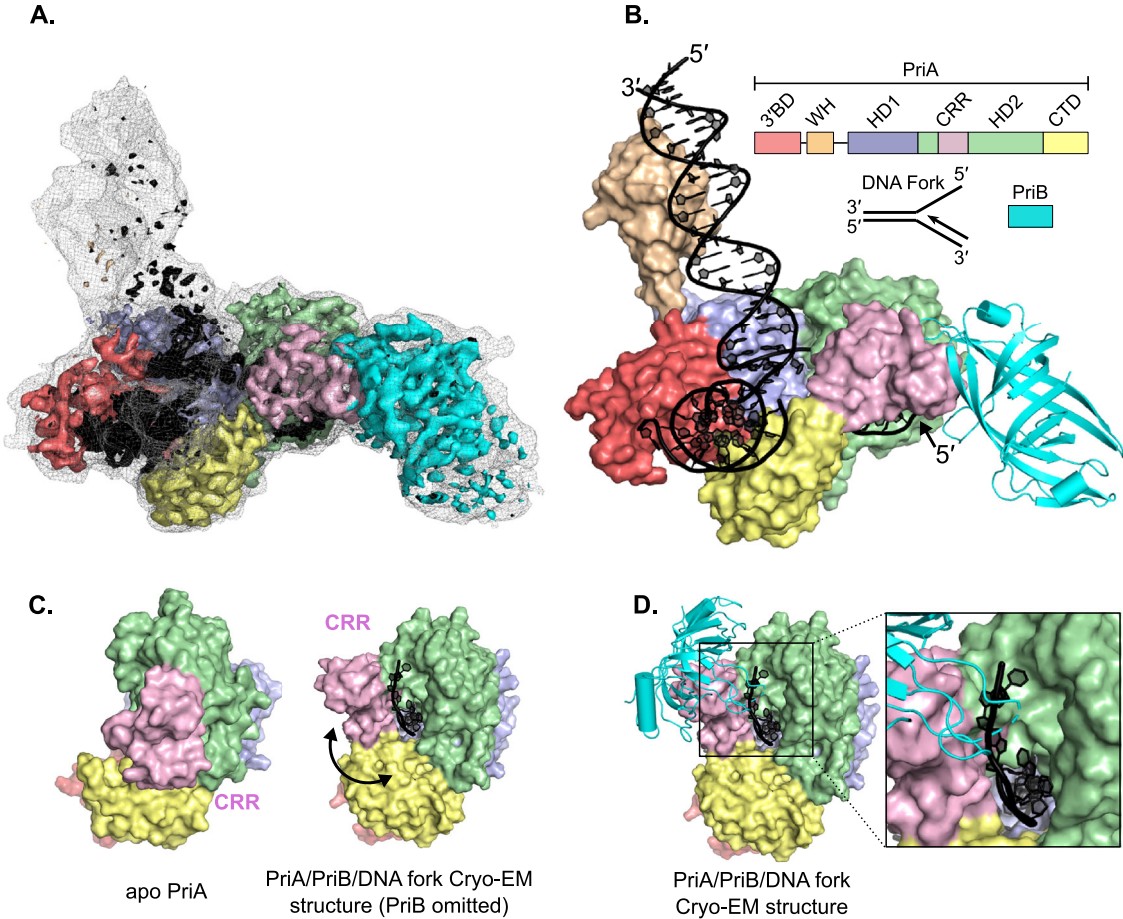

**Fig. 1 | A cryo-EM structure of PriA/PriB/replication fork. A** A 3.2 Å resolution cryo-EM density map (surface, colored to reflect modeled PriA and PriB domain positions within the density) overlayed onto a map low-pass filtered to 5 Å resolution (mesh). The filtered map highlights the extended DNA regions which are not visible at higher sharpening due to their inherent flexibility and thus lower resolution. **B** Cryo-EM structure of PriA (surface, colored by domain), PriB (ribbons, cyan), and replication fork DNA (black). A schematic of PriA (3'BD – 3'-Binding Domain, WH – Winged-Helix, HD1 and 2 – Helicase Domains 1 and 2, CRR – Cysteine Rich Region, CTD – C-terminal domain), PriB, and the synthetic replication fork are shown. **C** Comparison of the PriA-CRR position in apo PriA[24,27] (left) and the PriA/PriB/replication fork cryo-EM structure (right, PriB omitted). Arrow indicates rotation of the PriA-CRR in apo vs. DNA-bound structures. **D** Binding site of PriB on the rotated PriA-CRR. Inset shows close-up of ssDNA pore formed by PriA.

PriA (Fig. 1D). The DNA-binding surface of PriB is analogous to that used by the primary SSB in *E. coli*[33,42]. Since ssDNA binding activity by PriB is required for PriB binding to PriA/DNA complexes[32], the CRR and ssDNA likely combine to create the surface required for PriB binding.

### Additional observed classes of the *E. coli* PriA/PriB/replication fork complex

Two additional classes of PriA/PriB/replication fork structures were found by EM (Supplementary Fig. 1A and 3A-B). The first, accounting for ~15% of observed particles, is a dimer structure containing two PriA/replication fork complexes bound to either side of one PriB dimer ("Dimer 1", Supplementary Fig. 1A and 3A). Each individual PriA/PriB interface in Dimer 1 uses the PriA-CRR surface described above. Since replication fork DNA concentrations were stoichiometric with PriA in our cryo-EM sample preparations, this form likely reflects instances in which two replication-fork-bound PriA molecules stably associated with the same PriB homodimer. Given that abandoned DNA replication forks are present at a relatively low concentration in cells (occurring a small number of times per cell cycle[2–4]), this form may be an artifact of sample preparation. Possible functional relevance of Dimer 1 is examined below.

The second additional class (~5% of observed particles) was a "compact" dimer structure that consisted of two PriA/replication fork complexes bound to either side of two PriB dimers ("Dimer 2",

Supplementary Fig. 1A and 3B). In each PriB homodimer, one protomer was bound to the PriA-CRR binding site described above while the other protomer bound to a site on the PriA-CTD. The latter interface is formed by a β-hairpin from PriB abutting a loop in the PriA-CTD (Supplementary Fig. 3C). This secondary interface is not as well conserved as the CRR/PriB interface (Supplementary Fig. 3D) and is not predicted to form a stable interface by PISA[41]. Possible roles for the PriA-CTD/PriB interface are tested below.

### PriA positions PriB on the lagging strand ssDNA in solution

To begin analyzing the biochemical importance of interfaces observed in the EM structures, we first used site-specific crosslinking to test whether PriB is positioned on lagging strand ssDNA by PriA in solution as predicted by the cryo-EM structures. Our approach used PriB variants in which the unnatural amino acid *p*-benzoyl-*l*-phenylalanine (Bpa) is substituted for specific residues within the protein. Bpa is a UV-inducible crosslinker that reacts with C-H bonds within ~10 Å of the α-carbon, making it a useful probe for protein-nucleic acid interactions[43–45]. Bpa has been previously used to define PriA/replication fork interactions[27,40,46,47].

Our cryo-EM structures predict that PriA positions PriB on the lagging strand ssDNA ~10–15 nucleotides away from the replication fork junction. To test this prediction, a primer extension protocol was used to map crosslinking of two PriB Bpa variants (Leu16Bpa and

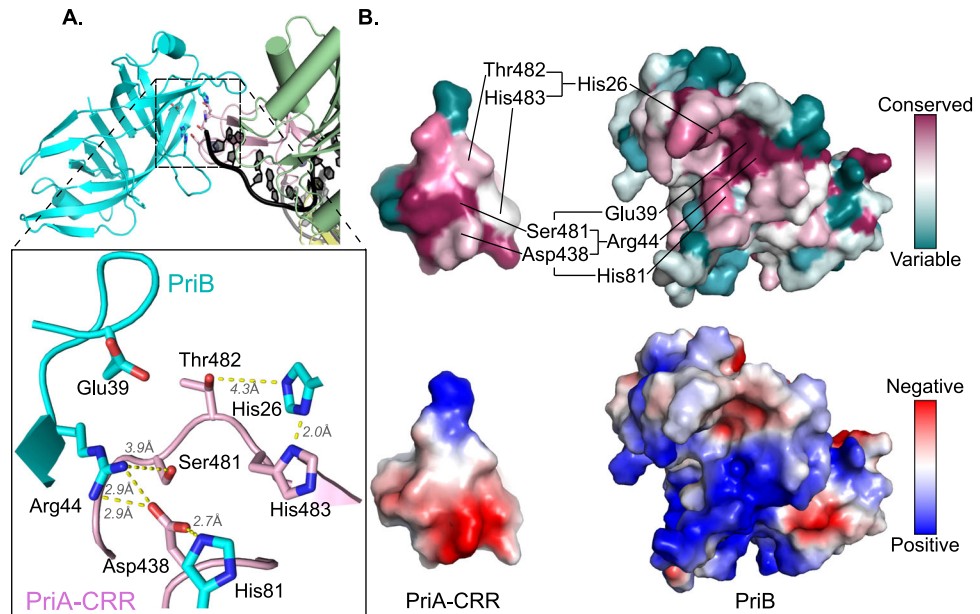

**Fig. 2 | The PriA-CRR/PriB binding interface. A** Interface formed by PriA-CRR (magenta) and PriB (cyan). Interatomic distances are labeled. **B** Surface representations of the PriA-CRR/PriB interface colored by conservation (top, calculated using the ConSurf server[72]) or electrostatic surface potential (bottom). Interacting residues are noted. Since PriB is not well conserved across bacteria[33], the calculation of PriA conservation was limited to 150 sequences from species containing *priB* genes.

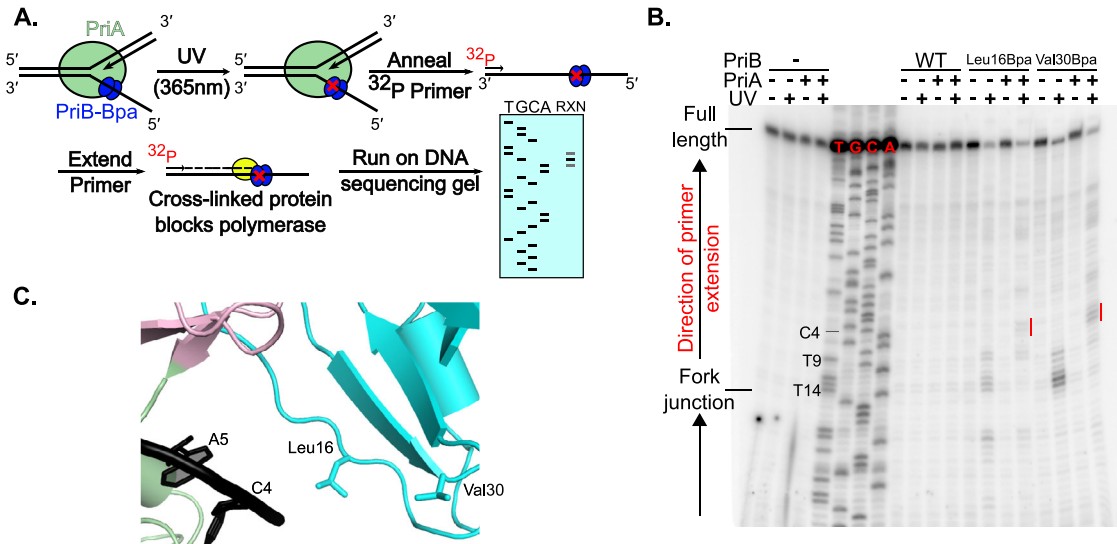

**Fig. 3 | Crosslinking affirms PriB/ssDNA interface in cryo-EM structure. A** Primer extension experiment schematic. A PriB Bpa variant is crosslinked to a replication fork in the presence or absence of PriA. A radiolabeled primer is annealed and extended by DNA polymerase until it is blocked by the crosslink. Primer resolution on a DNA sequencing gel reveals the location of the crosslink on the DNA strand. **B** Primer extension assay where Fork 3 (1 nM) was extended using oTW144

(Supplementary Table 1). Reactions containing PriB variants (200 nM, monomers) with or without PriA (2 nM) were exposed to UV prior to primer extension. Red bars show the location of PriA-dependent PriB crosslinks. The gel is a representative image of three replicate experiments. Source data are provided as a Source Data file. **C** Locations of PriB Leu16 and Val30 mapped onto the cryo-EM structure.

Val30Bpa) on lagging strand DNA. The Bpa variants retained the expected ssDNA binding affinity and specificity, indicating that the substitutions were well tolerated (Supplementary Fig. 4). For the primer extension assay, the PriB variants were crosslinked to the replication fork (with or without PriA), then a radiolabeled primer was annealed to the template lagging DNA strand. Polymerase extension of the primer proceeded until the polymerase collided with covalently linked PriB, generating a radiolabeled DNA strand with a length corresponding to the location of PriB on the replication fork. Comparison

of the primer extension products to DNA sequencing reactions facilitated mapping the crosslink site(s) onto the DNA sequence (Fig. 3A).

In the absence of PriA, PriB Leu16Bpa and Val30Bpa crosslink near the replication fork ssDNA-dsDNA junction with more minor crosslinking elsewhere along the lagging strand (Fig. 3B). Since Bpa displays some preference for crosslinking with thymine nucleobases[40,48], this reactivity is reflective of general ssDNA binding by PriB perhaps with a modest preference for binding at the fork junction. However, when PriA is present, PriB displays a marked decrease in crosslinking near

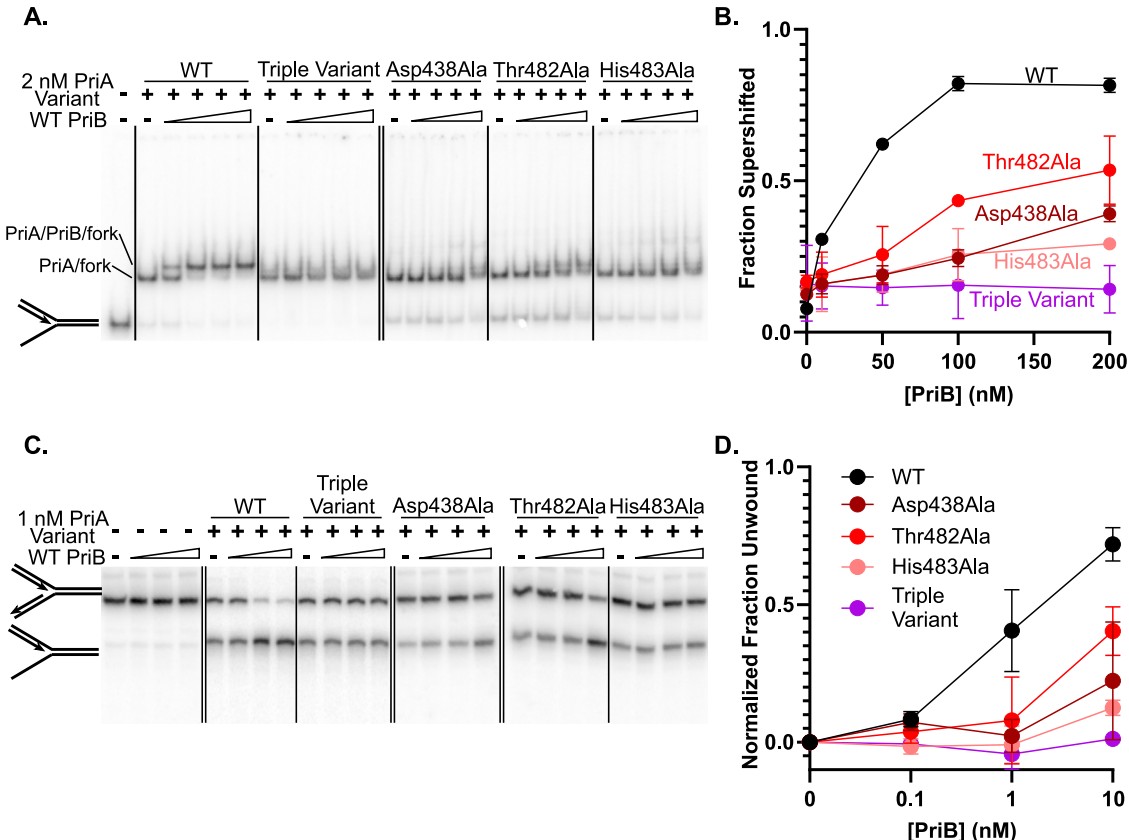

**Fig. 4 | PriA-CRR variants are defective in interacting with PriB. A** PriB inter-
action with PriA/replication fork complexes. Fork 2 (1 nM), PriA-CRR variant (2 nM),
and PriB (0, 10, 50, 100, or 200 nM, monomers) were co-incubated and resolved
using native TBE-PAGE. **B** Quantification of the ratio of PriB-supershifted (PriA/PriB/
DNA complex) to PriA-shifted (PriA/DNA complex) bands in (**A**). **C** PriB stimulation
of PriA-CRR variant helicase activity. Fork 2 (1 nM), PriA-CRR variants (1 nM), and
PriB (0, 0.1, 1, 10 nM, monomers) were incubated with ATP. DNA unwinding pro-
ducts were resolved using TBE-PAGE. **D** Quantification of percentage unwound
normalized to 0 nM PriB reactions in (**C**). Data points are the mean of three mea-
surements +/− standard deviation. Source data are provided as a Source Data file.

the fork junction and new primer extension bands appear 10-12
nucleotides away from the fork junction, most notably for the
Val30Bpa PriB variant (Fig. 3B). This PriA-dependent positioning aligns
with two features of the EM structures. First, PriA exclusion of PriB
from 8-9 junction-proximal nucleotides of the lagging strand is con-
sistent with PriA sequestering this region in the pore. Second, PriB
Leu16 and Val30 are adjacent to the C4 base in the ssDNA region within
the EM structure, consistent with their observed crosslinking to
nucleotides 10-12 in the primer extension assay (Fig. 3C).

**Formation of the PriA/PriB/replication fork complex requires
the PriB docking site on the PriA-CRR and a single PriA binding
site on PriB**
The PriA/PriB/replication fork structures identified two possible PriB-
binding interfaces and three possible quaternary arrangements of PriA
and PriB (Fig. 1 and Supplementary Fig. 3A, B). To test the possible PriB
docking sites on PriA biochemically, we purified a panel of PriA variants
that substitute Ala for residues in the PriA-CRR (Asp483, Thr482, and/
or His483 (Fig. 2)) or CTD (Arg657, Trp671, or Leu673 (Dimer 2, Sup-
plementary Fig. 3B, D)). We reasoned that if the CRR and/or CTD
binding surface is important for formation of PriA/PriB complexes on
DNA replication forks or PriB stimulation of PriA activity, these
sequence changes would impair PriB activity with the PriA variants.

Linked dimer (LD) PriB variants were also purified to test the
possibility that PriA binding to both PriB protomers is required for
PriA/PriB activity (Dimer 1 structure, Supplementary Fig. 3A). For these
variants, the C-terminus of one PriB monomer was directly fused to the
N-terminus of a second PriB monomer in a single polypeptide. The

termini are adjacent to one another in crystal structures of PriB[35-38],
making a simple fusion possible. Wild-type LD (WT LD) PriB was pur-
ified along with three LD variants in which Arg44, a residue that is
important for *E. coli* PriB binding to PriA[29], was changed to Ala in either
the N-terminal (N-term Arg44 Ala LD), the C-terminal (C-term Arg44
Ala LD), or both (double Arg44 Ala LD) PriB domains. If both PriA-
binding sites on PriB are necessary for an active PriA/PriB/replication
fork structure as suggested by the Dimer 1 cryo-EM structure, then
only the WT LD PriB variant should bind PriA and stimulate its activity.
If, instead, only one PriA-binding site is necessary, then the WT LD,
N-term Arg44Ala LD, and C-term Arg44Ala LD should function, with
only the double Arg44Ala PriB LD variant being unable to bind or
stimulate PriA.

We first tested the PriA variants' ability to recruit PriB to replica-
tion forks using an electrophoretic mobility shift assay (EMSA). As has
been observed previously[28,32,47], binding by *E. coli* PriA reduced the gel
mobility of radiolabeled replication fork DNA, and PriB binding to the
PriA/DNA complex further reduced mobility in a concentration-
dependent manner (Fig. 4A). Each of the PriA variants shifted replica-
tion fork mobility similarly to wild-type PriA (WT shifts 91.9 ± 4.1% of
the labeled fork, Asp438Ala shifts 95.0 ± 1.9%, Thr482Ala shifts
91.3 ± 6.5%, His483Ala shifts 92.2 ± 3.7%, and the triple variant shifts
95.8 ± 3.1%), indicating that each retained DNA binding activity (Fig. 4A
for CRR variants and Supplementary Fig. 3E for CTD variants). How-
ever, differences in PriB association with the PriA-CRR variants were
apparent in the PriB titrations (Fig. 4A, B). The Asp438Ala and
His483Ala single variants were strongly impaired for PriB binding to
the PriA/DNA complexes while the Thr482Ala PriA variant had a lesser

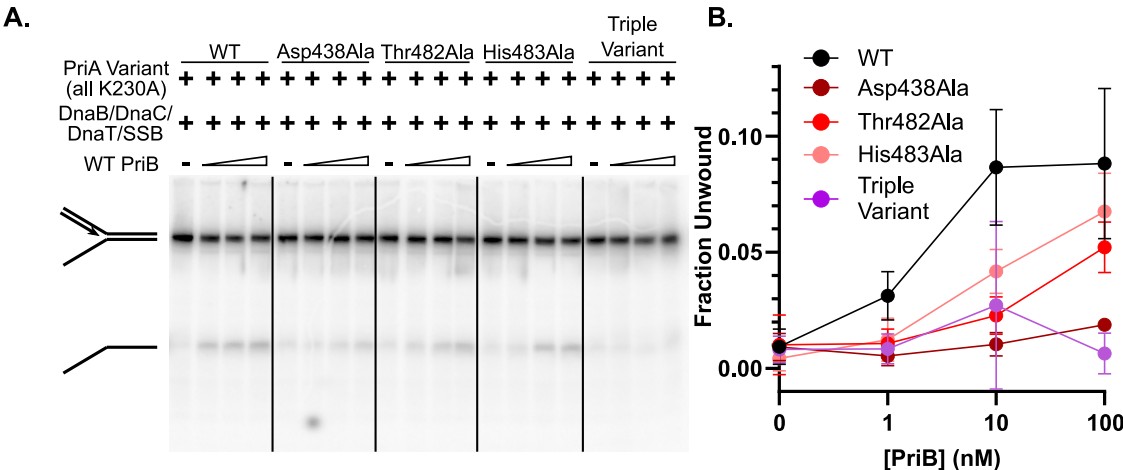

**Fig. 5 | PriA-CRR variants are defective in loading DnaB onto replication forks.** **A** Fork 2 (1 nM) was incubated with DnaB (240 nM, monomers), DnaC (200 nM, monomers), PriA K230A variants (10 nM), DnaT (960 nM), and SSB (120 nM) in the presence of PriB (0, 1, 10, 100 nM, monomers). DNA unwinding products were resolved using TBE-PAGE. **B** Quantification of percentage unwound as a function of PriB concentration. Data points are the mean of three measurements +/− standard deviation. Source data are provided as a Source Data file.

impact. The PriA-CRR triple variant, which combined all three residue changes, was unaffected by the addition of PriB, reflecting the greatest loss of PriB binding. The binding results for wild-type PriA and the triple CRR variant were further confirmed using size-exclusion chromatography (Supplementary Fig. 5). In contrast, PriB binding by the PriA-CTD variants was unaffected by the sequence changes (Supplementary Fig. 3E, G). These results define a major role for the PriA-CRR in PriB binding whereas the possible PriA-CTD interface from the Dimer 2 structure appears to be dispensable for PriB binding to PriA/DNA complexes.

The WT LD and single Arg44Ala LD PriB variants bound PriA/replication fork complexes as well as wild-type PriB (Supplementary Fig. 3I, K). However, the double Arg44Ala LD variant was deficient in binding, mirroring the result with Arg44Ala PriB. This indicates that PriB needs only one active binding surface to dock onto PriA/replication fork complexes.

### PriB stimulation of PriA DNA helicase activity requires the PriB docking site on the PriA-CRR and a single PriA binding site on PriB
We next measured PriB stimulation of DNA unwinding by the PriA variants. PriA selectively unwinds dsDNA lagging strands in replication forks (Fig. 4C). In the absence of PriB, each of the PriA-CRR and PriA-CTD variants unwound DNA with similar efficiencies to that of wild-type PriA (Fig. 4C and Supplementary Fig. 3F and 6A, C). As has been observed previously[32], PriB stimulated wild-type PriA helicase activity in a PriB concentration-dependent manner (Fig. 4C, D). However, PriB titrations revealed differences in the ability of the PriA variants to be stimulated. Each of the PriA-CRR variants was defective in PriB stimulation, as evidenced by the requirement for higher PriB concentrations being needed to stimulate DNA unwinding. The Asp438Ala and His483Ala single variants displayed the strongest individual defects, whereas the Thr482Ala sequence change led to a more modest impact on PriB stimulation (Fig. 4C, D), paralleling the PriB binding results from above. The PriA-CRR triple variant had the most dramatic impact, with no detectable increase in substrate unwinding at any concentration of PriB tested. In contrast, PriB stimulation of the PriA-CTD variants was indistinguishable from that of wild-type PriA (Supplementary Fig. 3F, H). These results again emphasize the importance of the CRR binding site for PriB stimulation of PriA biochemical function.

In agreement with their ability to bind PriA/DNA complexes, the WT LD and single Arg44Ala LD PriB variants stimulated PriA DNA unwinding as well as wild-type PriB (Supplementary Fig. 3J, L). However, the double Arg44Ala LD variant stimulated ~100-fold more weakly than wild-type PriB, mirroring the result with Arg44Ala PriB. This finding indicates that PriB requires only one active binding surface to fully stimulate PriA helicase activity.

To examine the specificity of the effects of the PriA variants, we tested whether they retained SSB stimulation of helicase activity. SSB interacts directly with PriA at a site that is distinct from the PriB-binding interface(s) and stimulates helicase activity[24,49]. Helicase activity measured in the presence and absence of SSB showed that wild-type PriA and all of the PriA interface variants (CRR and CTD) were stimulated similarly by SSB (Supplementary Fig. 6B, D), showing that the PriA-CRR variants are exclusively defective in stimulation by PriB.

Since the PriA-CTD variants retain full activity in binding and stimulating PriA, the PriA-CTD/PriB interface found in the Dimer 2 cryo-EM structure is most likely an artifact of sample preparation. Similarly, PriB LD variants with only one PriA binding site function normally, suggesting that the Dimer 1 cryo-EM structure is also an artifact. Further biochemical and genetic analysis were therefore limited to the PriA-CRR variants.

### PriA/PriB-mediated DnaB reloading requires the PriB docking site on the PriA-CRR
PriA, PriB, and DnaT function together to recruit the DnaB/DnaC helicase/loader to abandoned DNA replication forks and to reload DnaB in a reaction that can be recapitulated in vitro[9,50]. In one version of the assay, an ATPase-inactive PriA variant (Lys230Ala)[51] is incubated with a radiolabeled replication fork, PriB, DnaT, DnaC, DnaB, and SSB. DnaB loading by the PriA/PriB/DnaT complex results in unwinding of the parental dsDNA. Use of the Lys230Ala PriA variant simplifies interpretation of the assay result (parental dsDNA unwinding) since DnaB is the only active helicase in the reaction[9].

To determine whether interaction between the PriA-CRR and PriB is important for DnaB loading, we measured the impact of the PriA-CRR variants in this assay. In agreement with prior observations[9,29], DnaB loading by PriA/PriB/DnaT and subsequent DnaB-mediated unwinding of the parental dsDNA of the replication fork is stimulated in a PriB concentration-dependent manner, unwinding ~10% of the substrate at the highest PriB concentration tested (previous iterations of this assay have reported <20% unwinding efficiency[9,29,50], likely due to complexity of the reconstituted system) (Fig. 5). Among the single point variants, PriA Asp483Ala was most significantly

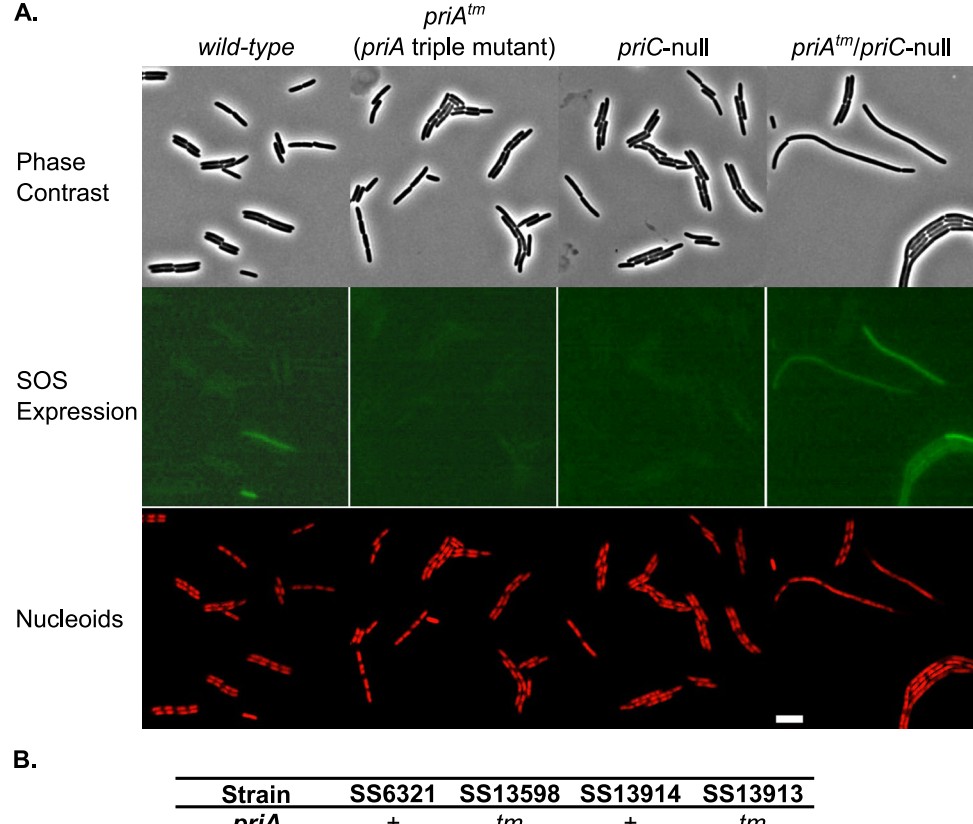

**Fig. 6 | Mutation of the PriA-CRR/PriB interface causes a replication restart-deficiency phenotype in *E. coli* when paired with a *priC*-null mutation.**
**A** Representative images of cells carrying the *wild-type, priA^tm, priC*-null, *or priA^tm/priC*-null alleles (top: phase contrast, middle: SOS expression, bottom: nucleoids). Scale bar = 5 μm. **B** Quantification of cell area and SOS expression along with qualitative nucleoid partitioning phenotypes. Approximately 700–1500 cells were counted for quantification.

reduced in its ability to drive DnaB loading, as evidenced by low levels of DNA unwinding by DnaB in all PriB concentrations tested. The PriA His483Ala and Thr482Ala variants required higher PriB concentrations to catalyze DnaB loading than wild-type PriA in the assay, consistent with reduced PriA/PriB complex formation. As with our prior PriB binding and helicase results, no DnaB loading activity was observed for the PriA-CRR triple variant at any PriB concentration tested. Thus, the PriA-CRR/PriB interface appears important for loading DnaB onto DNA replication forks.

**Disrupting the PriA-CRR/PriB interface causes DNA replication defects in vivo**
To measure the importance of the PriA-CRR/PriB interface in cells, the impact of integrating the CRR triple mutation into the *priA* gene in *E. coli* was assessed (*priA^tm* - allele *priA3SS::Asp438Ala, Thr482Ala, His483Ala*). We predicted that the *priA^tm* mutation would selectively disrupt the PriA-PriB pathway. Cells with a full *priB* deletion (Δ*priB* - allele *del(priB)302*), which eliminates the PriA-PriB pathway, maintain intact PriC-dependent replication restart pathways and do not display replication restart deficiency phenotypes[52]. Accordingly, *priA^tm* mutant cells were indistinguishable from wild-type cells with cell morphologies indicative of normal growth (Fig. 6, Supplementary Table 3). We

further predicted that restricting in vivo replication restart solely to the PriA-PriB pathway by combining a *priC*-null mutation (*priC303::-kan*) with the *priA^tm* mutation would lead to genome maintenance defects. As predicted, *priA^tm/priC*-null cells are highly filamentous, consistent with genomic stress and induction of the SOS DNA damage response (Fig. 6 and Supplementary Table 3). We note that the phenotypes are not as drastic as has been observed for Δ*priB/priC*-null or *priA*-null mutants[53], indicating that some modest replication restart activities are maintained in *priA^tm/priC*-null cells. Controls pairing Δ*priB* with *priA^tm* show no cellular defects, indicating that the phenotype is specifically linked to the *priA^tm* mutation impairing the PriA-PriB replication restart pathway (Supplementary Fig. 7 and Supplementary Table 3). The modest replication restart activity may be due to residual binding of PriB to the PriA-CRR triple variant in vivo that is not detected in vitro. It is possible that SSB bound to the lagging strand in cells recruits PriB to the replication fork through the known SSB-PriB interaction[54]. Since the lagging strand in cells can be >1000 bases long with many bound SSBs, the local concentration of PriB associated with SSB at the replication fork may be higher in cells than in the in vitro assays that have been used here. Such a phenomenon could bolster a weak interaction between PriB and the PriA-CRR triple variant, allowing modest PriA-PriB pathway activity in *priA^tm* cells.

To further study the cellular defects resulting from this combination of alleles, the strains were engineered to encode a HupA:mCherry fusion protein to assess nucleoid partitioning in cells and a *gfp* gene driven by the SOS-inducible *sulA* promoter to measure SOS induction[7,52]. Chromosomes in the strains containing wild-type *priA*, *priA^tm*, *priC*-null, Δ*priB*, or *priA^tm*/Δ*priB* partition normally (Fig. 6 and Supplementary Fig. 7), indicative of normal DNA replication and segregation of daughter chromosomes. These strains also have low GFP levels, indicating low DNA damage levels (Fig. 6, Supplementary Fig. 7, and Supplementary Table 4). In contrast, chromosomes in *priA^tm*/*priC*-null cells display partitioning defects and have significantly increased GFP signal, both indicating elevated genomic instability (Fig. 6 and Supplementary Table 4). These results demonstrate that the PriA-CRR/PriB interface is critical for activity in vivo and that disrupting this interaction hinders the PriA-PriB replication restart pathway in cells.

## Discussion

Bacterial DNA replication restart pathways reload replisomes onto prematurely terminated DNA replication forks and D-loops formed during dsDNA break repair. These pathways are initiated by robust coupling of structure-specific replication fork binding to assembly of the protein complexes required to reload the replicative helicase. To better understand how the PriA-PriB replication restart pathway functions, we examined the PriA/PriB/replication fork complex structure using cryo-EM single-particle analysis. The resulting 3.2 Å global resolution cryo-EM structure of the complex revealed an extensive interface formed between PriA and each arm of the DNA replication fork that confers exquisite structural specificity. Replication fork binding triggered a dramatic rearrangement of the PriA structure to create a pore encircling lagging strand ssDNA while simultaneously exposing a docking site for PriB. Together with the structure, biochemical and genetic studies define a switch-like mechanism for replication restart initiation in which PriA domain rearrangement directly couples replication fork recognition to PriA/PriB complex formation to ensure robust, high-fidelity replication re-initiation.

The network of protein/DNA interactions found in the PriA/PriB/replication fork complex structure included extensive PriA domain interactions with the leading strand duplex (3′BD and CTD), the parental DNA duplex (WH and HD1), and the single-stranded lagging strand (3′BD, HD1, HD2, CRR, and CTD, along with PriB). Interactions with the lagging strand were the most numerous, with 42 residues cooperating to bind the phosphates, sugars, and/or bases of 14 nucleotides of ssDNA. Dense protein interactions with lagging strand ssDNA suggests that its sequestration in the complex could be important for a step such as DnaB reloading, which requires ~20 nucleotides of ssDNA[55,56]. PriA encircles 8-9 nucleotides of lagging strand ssDNA within the structure and PriB binds additional bases extending from the 5′ end via its DNA-binding surface (site size is ~14 bases/dimer)[35]. Thus, the length of ssDNA bound to PriA/PriB correlates well with the loading site size for DnaB. DnaT, which joins the complex after PriB docks onto PriA and competes with PriB for ssDNA binding[28,29,57,58], could rearrange the PriA/PriB complex in a manner that exposes ssDNA for DnaB loading and could potentially aid in DnaB recruitment. Alternatively, DnaB may be loaded at a site outside of the PriA/PriB complex and DnaB translocation or displacement activity could be important for dislodging the replication restart proteins after replication re-initiation.

The ssDNA lagging strand used in our replication fork mimics an abandoned DNA replication fork where lagging strand priming has not occurred close to the fork junction. The substrate also mimics D-loop structures that arise from recombinational repair, which are known to rely on the PriA-PriB pathway for processing[23,52]. Since PriA ATPase and helicase activity are not required for PriB-PriB dependent replication restart in vivo[59], similar replication forks to that used in our cryo-EM structure are likely to be common substrates for PriA-PriB-dependent

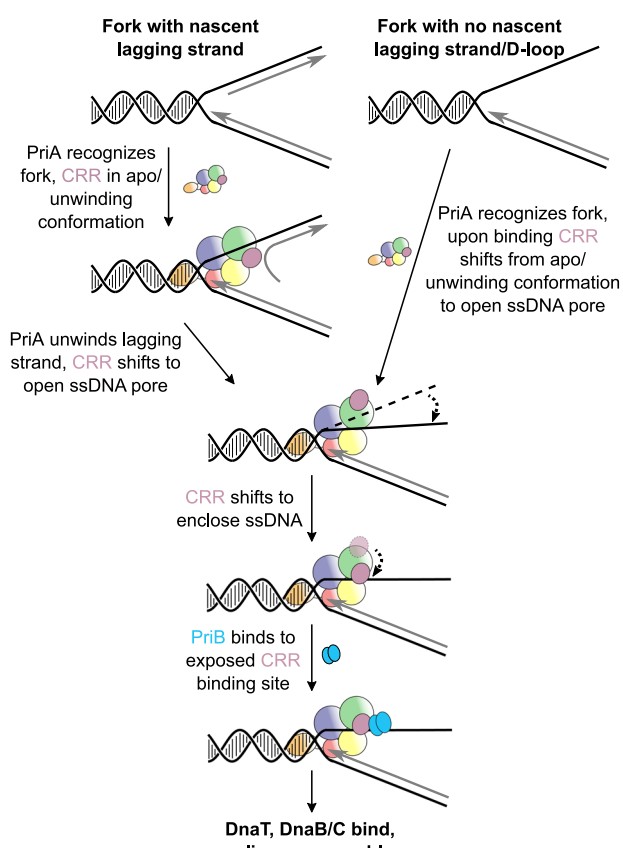

**Fig. 7 | Recognition of forked DNA structures regulates replication restart via conformational changes in PriA.** PriA recognizes forked DNA substrates with a dsDNA lagging strand (left) or a ssDNA lagging strand (right). When dsDNA is present, the PriA-CRR is positioned to unwind the DNA until ssDNA has been exposed, which triggers a conformational rearrangement of PriA that forms the ssDNA pore and exposes the PriB binding site. Passage of ssDNA into the PriA requires significant movement of the CRR. PriB subsequently binds, triggering recruitment of DnaT and DnaB/C, ultimately leading to replisome assembly.

replication restart in *E. coli*. One outstanding question, however, is how does ssDNA enter the PriA pore? Since the chromosome is circular in *E. coli*, the lagging strand DNA lacks a free end and cannot thread into a preformed pore in PriA. Instead, it appears that the PriA structure must crack open to allow the lagging strand to pass into the PriA pore (Fig. 7). Such a loading mechanism would require the CRR to move away from the positions observed in the EM and prior crystal structures to create an opening wide enough to allow ssDNA passage. The mechanism that would trigger a change of this magnitude is not known, but it could potentially rely on ssDNA binding to positions with PriA to allosterically modulate the position of the CRR.

In addition to its role in forming a ssDNA binding pore and binding PriB, the PriA-CRR is critical for helicase activity, with a β-hairpin element within the domain being essential for DNA unwinding[47,60]. The β-hairpin, which is exposed in a channel in apo PriA, has been proposed to bind the ss/dsDNA junction and act as a wedge to unwind DNA[24,47]. Consistent with this model, residues from the tip of the β-hairpin and other residues in the channel crosslink to lagging strand dsDNA[40,47]. However, rearrangement of the PriA-CRR in the cryo-EM structure closes the dsDNA-binding channel and tucks the β-hairpin against the PriA-HD2, making it inaccessible to DNA. How then does PriA unwind the lagging strand in substrates with dsDNA lagging strands? We propose that, when bound to a dsDNA substrate, the PriA-CRR is positioned similarly to observed apo structures with the β-hairpin available to act as a DNA strand-separation wedge. Once

PriA has unwound sufficient ssDNA on the lagging strand, we hypothesize that the CRR is triggered to open, allowing ssDNA to enter the pore (Fig. 7). This movement also exposes the PriB binding interface observed in our cryo-EM structure. The model therefore predicts that the conformational state of the PriA-CRR will differ depending on structure (dsDNA or ssDNA) of the lagging strand. Further DNA unwinding by the PriA/PriB complex may rely on PriB to function as the DNA unwinding wedge. Since repositioning of the PriA-CRR leads to PriB docking (which stimulates DnaT binding[28,29]), structure-specific CRR movement represents a parsimonious mechanism that allows PriA to recognize and process different DNA replication fork structures while selectively triggering replication restart at appropriate sites. Similar strategies may also have been adapted by other proteins that bind and process stalled replication forks in archaeal and eukaryotic systems.

## Methods

### Plasmid construction

**PriA variants.** Site-directed mutagenesis was performed on a PriA- or PriA-K230A overexpression plasmid[29] to encode PriA variants. Open reading frames of all mutant plasmids were confirmed by sequencing.

**PriB Bpa variants.** Site-directed mutagenesis was performed on a PriB-overexpression plasmid[37] to place a stop codon (TAG) at the specified site of Bpa substitution. Open reading frames of all mutant plasmids were confirmed by sequencing.

**PriB linked dimer.** A gene encoding a linked dimer of PriB was synthesized by Integrated DNA Technologies. The open reading frame included one wild-type *priB* sequence followed immediately by a codon-optimized *priB* sequence. The gene was subcloned into a pET29b expression vector. Site-directed mutagenesis was performed on the linked-dimer overexpression to plasmid to create PriB linked-dimer variants. Open reading frames of all mutant plasmids were confirmed by sequencing.

**Protein purification.** *E. coli* PriA and PriA variants (containing an N-terminal His tag) were expressed and purified as described previously[46,61].

BL21(DE3) *E. coli* cells transformed with an overexpression plasmid for PriB or PriB variant were grown at 37 °C in Luria Broth supplemented with 50 μg/mL each of kanamycin and chloramphenicol to an $OD_{600 nm}$ of ~0.6 were induced to express PriB with 1 mM isopropyl β-D-1-thiogalactopyranoside for four hours. Cells were pelleted and resuspended in 40 mL Lysis Buffer (20 mM Tris-HCl, pH 7, 10% glycerol, 1 M NaCl, 10 mM imidazole, 1 mM β-mercaptoethanol, 1 mM phenylmethylsulfonyl fluoride, 1 mM benzamidine, 1 EDTA-free protease inhibitor tablet), lysed by sonication, and clarified by centrifugation. All purification steps were carried out using an ÄKTA Pure FPLC system (GE) at 4 °C. The soluble lysate was loaded directly onto a 5 mL HisTrap FF crude FPLC column (GE) equilibrated in Equilibration Buffer (Lysis Buffer lacking protease inhibitors). Note that untagged *E. coli* PriB binds to Nickel resin. The column was washed with 50 mL Equilibration Buffer and eluted with 30 mL Elution Buffer (20 mM Tris-HCl, pH 7, 10% glycerol, 1 M NaCl, 250 mM imidazole, 1 mM β-mercaptoethanol). The eluate was diluted with 70 mL Dilution Buffer (10 mM HEPES pH 7, 10% glycerol, 1 mM β-mercaptoethanol) then loaded directly onto a 20 mL HiPrep sulfopropyl sepharose fast flow (SPFF) FPLC column (GE) and eluted using a 100–1000 mM NaCl gradient in SPFF Buffer (10 mM HEPES-HCl, pH 7.0, 10% glycerol, 100–1000 mM NaCl, 10 mM dithiothreitol). Eluted protein was concentrated and loaded onto a HiPrep S100 FPLC column (GE) equilibrated in 1 M NaCl SPFF Buffer. Purified protein was concentrated, dialyzed overnight against storage buffer (10 mM HEPES-HCl, pH 7.0, 50% glycerol, 500 mM NaCl, 10 mM dithiothreitol), and stored at −20 °C.

PriB Bpa variants were expressed in the same manner as previously reported for PriA Bpa variants[46] and purified like wild-type PriB. PriB linked-dimer variants were expressed and purified like wild-type PriB. PriB linked-dimers purify as monomers that resolve at twice the wild-type mass on a denaturing PAGE gel but have the same apparent molecular weight as wild-type PriB on size-exclusion chromatography.

**Construction of synthetic DNA replication fork substrates.** The synthetic DNA replication fork used in EM (Fork 5) was generated by combining 20 μM each of oAD027-029 (Supplementary Table 1), heating to 95 °C for 10 min, and slowly cooling to 4 °C over several hours in a thermal cycler. The substrates were then resolved through a 15% TBE-PAGE gel, located via UV shadowing, and excised. The gel slices were shredded and placed in 1X TE buffer overnight at 4 °C to diffuse. The DNA-containing buffer was extracted and ethanol precipitated to purify and concentrate the substrates. The final product was resuspended in 1X TE buffer and stored at 4 °C.

Radiolabeled DNA replication fork substrates (Forks 1-4) were constructed as described previously[46,62,63]. Briefly, oligonucleotide 3L-98 or oTW141 was 5′-radiolabeled with ATP-[γ-³²P] using T4 polynucleotide kinase, annealed to their respective partners (see Supplementary Table 1), and purified via PAGE electrophoresis. In the case of the primer extension experiment, the fork substrate was unlabeled and the primer extension primer, oTW144, was 5′-radiolabeled.

**Cryo-EM sample preparation and imaging.** 10 nmol PriA, 10 nmol Fork 5, and 20 nmol PriB (monomers) were co-incubated in 5 mL S200 Buffer (50 mM Tris-HCl, pH 8, 2 mM dithiothreitol, 5 mM ethylenediaminetetraacetic acid, 75 mM NaCl) on ice and concentrated to 500 μL. The complex was purified on a Superdex 200 Increase 10/300 GL analytical size exclusion FPLC column (Cytiva) in S200 Buffer. Peak fractions were combined and concentrated to ~1 mg/mL. Samples were applied to Ultrafoil 1.2/1.3 grids (Quantifoil) that had been glow-discharged for 30 s using a GloQube Plus glow discharge system (Qurom Inc). The grids were plunge frozen at 4 °C using a Vitrobot Mark VI (ThermoFisher) under 100% humidity with a blot time of 4 s. Movies were collected at 300 kV on a Titan Krios controlled by SerialEM[64]. 1595 movies were collected using a Gatan K3 camera operating in CDS mode and BioQuantum energy filter with a 20 eV slit width at a calibrated pixel size of 1.079 Å. In a separate session 1137 movies were collected using a Falcon 3 camera operating in counting mode at a calibrated pixel size of 0.8330 Å. The K3 dataset was collected with a total dose of 100 e/Å² split into 102 fractions and the Falcon 3 dataset was collected with a total dose of 60 e/Å² split into 60 fractions. Images were collected with a defocus range of 0.5 to 2.5 μm.

**Cryo-EM data processing.** 2732 movies were imported into cisTEM 2.0[65] for data processing following the standard cisTEM processing workflow. After motion correction and contrast transfer function (CTF) parameter estimation, only micrographs with a detected CTF fit resolution better than 4.0 Å were kept for further processing (1925 movies were selected). Initially ~900,000 particles were automatically picked using the cisTEM disc picker and subjected to 2D classification. Of these, ~250,000 particles were classified into good class averages with high resolution features. Particles were further classified into three subsets, corresponding to the PriA/B complex monomer (~190 k particles) and two dimer forms (Dimer 1 ~ 40 k and Dimer 2 ~ 7 k particles) manually based on the class averages and further processed independently. Ab initio reconstructions were created from the best selected class averages and further refined via auto-refinement imposing C1 symmetry for the monomer set and C2 symmetry for the dimer sets (see Supplementary Fig. 1A for workflow overview).

Both dimer forms exhibited strong preferential orientation and despite relatively high global resolutions as measured by the Fourier shell correlation (FSC) and 0.143 cut-off (3.8 Å and 3.9 Å) the overall quality of the maps is quite poor with features at resolutions higher than ~7 Å resolution only visible in one specific orientation. Given the poor quality of the maps and the fact that at the obtained resolutions they both appear to be two copies of the monomer form, further investigation was restricted to the monomer form.

The monomer form refinement led to a map with a global resolution of 3.2 Å as measured by the FSC and 0.143 cut-off. This map exhibited good density for most of PriA, the core DNA, and PriA/PriB interface, facilitating direct model building (see Supplementary Fig. 1B). As expected, the extended DNA strands and PriA-WH demonstrated a high degree of flexibility and are essentially not visible in the final high resolution sharpened map. In addition, the PriB protomer not interacting with PriA appears highly heterogeneous and exhibits poor density. To improve the density of the parental DNA strand and PriA-WH, masked focused classification was performed on this region, resulting in a class with density of sufficient quality to rigid body fit the parental DNA extending away from PriA and the PriA-WH (Supplementary Fig. 1C).

**Model building, refinement, and validation.** Previously solved structures for PriA (PDB: 4NL4[24] and 6DCR[27]) and PriB (PDB: 1TXY[37]) were rigid body fit into the density using UCSF ChimeraX[66]. The core DNA model was built manually and the entire model was edited using COOT[67] and refined using PHENIX real space refinement[68]. The modest resolution of the maps for the PriA-WH and parental duplex DNA limited modelling to low-confidence rigid-body fitting of these elements and they are included in figures for illustrative purposes but are not present in the final model. Uncertainty in this region further limited determination of the register of the lagging strand ssDNA. Validations were performed using PHENIX. Model visualization and figures were generated using ChimeraX or COOT. The cryo-EM map for PriA/PriB/DNA was deposited in the Electron Microscopy Data Bank (EMD-28959) and the final model was deposited in the Protein Data Bank (8FAK). Dimer 1 and Dimer 2 structures were assembled by fitting the refined PriA/PriB/DNA structure into EM density and were not refined further due to limitations in the resolution.

**PriB Bpa crosslinking experiments.** PriB Bpa variants (200 nM, monomers) were incubated with 1 nM radiolabeled Fork 3 in 50 mM Tris-HCl, pH 8, 0.1 mg/mL BSA, 1 mM dithiothreitol, 5 mM ethylenediaminetetraacetic acid, 6% glycerol on ice for 15 min. Wild-type PriA (2 nM) was present where indicated. Reactions were exposed to 365 nm UV light for five minutes at room temperature and then rested on ice for two minutes. A final UV exposure was performed, on ice, for 15 min. SDS buffer was added to denature the protein and the reactions were resolved in a 7.5% TBE-PAGE gel. The experiment was performed in triplicate.

To determine the DNA strands to which PriB was crosslinking, the above protocol was repeated with variants of Fork 3 that each had a different strand radiolabeled. Boiling these substrates after crosslinking allows for tracking of crosslinking to individual strands.

Primer extension experiment with PriB Bpa variants has been described previously[40,46]. Here, the protocol is the same except 1 nM unlabeled Fork 3 was incubated with 200 nM PriB Bpa variant (monomers). Wild-type PriA (2 nM) was included where indicated.

**Electrophoretic mobility shift assays.** PriA and PriA/PriB EMSAs have been described previously[32]. Briefly, 1 nM Fork 1 was incubated with 2 nM PriA variant and 0, 10, 50, 100, or 200 nM PriB variant (monomers) in 50 mM Tris-HCl, pH 8, 0.1 mg/mL bovine serum albumin, 1 mM dithiothreitol, 5 mM ethylenediaminetetraacetic acid, 6% glycerol on ice for 15 min. The reactions were then resolved through a 4%

TBE-PAGE gel. As the PriB linked-dimer variants' molecular weight is doubled, they were included at 0, 5, 25, 50, or 100 nM. Gels were quantified using the ImageQuant software and analyzed with the GraphPad Prism software.

**Helicase assays.** PriA activity assays were performed by incubating 1 nM Fork 2 with 0, 0.1, 1, 2, or 5 nM PriA variant in 50 mM HEPES-HCl, pH 8, 40 mg/L bovine serum albumin, 2 mM dithiothreitol, 4 mM magnesium acetate, and 2 mM ATP at 37 °C for 10 min. Reactions were terminated by adding 20 mM ethylenediaminetetraacetic acid, 0.5% SDS, 0.2 g/L proteinase K and 2.5 ng/μL unlabeled 3L-98 oligo, and incubating at 37 °C for 20 min. DNA loading dye was added and the samples were resolved using a 7.5% TBE-PAGE gel. Gels were quantified using the ImageQuant software and analyzed with the GraphPad Prism software.

For PriB stimulation assays, the reactions were set up as above except with 1 nM Fork 2, 1 nM PriA variant, and 0, 0.1, 1, or 10 nM PriB variant (monomers). PriB linked-dimer variants were included at 0, 0.05, 0.5, or 5 nM (monomers).

For SSB stimulation assays, the reactions were set up as for the PriB stimulation assays except with 0, 100, 250, or 400 nM SSB (monomers).

**DnaB reloading assays.** DnaB loading assays were performed as previously described[9]. Briefly, 1 nM Fork 2, 240 nM DnaB, 200 nM DnaC, 10 nM PriA K230A variant, 960 nM DnaT, and 120 nM SSB were incubated in 50 mM HEPES-KOH, pH 8, 40 μg/mL bovine serum albumin, 2 mM dithiothreitol, 2 mM ATP, 4 mM magnesium acetate for 30 min at 37 °C. Wild-type PriB was included at 1, 10, or 100 nM where indicated. The reactions were terminated with 20 mM ethylenediaminetetraacetic acid, 0.5% SDS, 0.2 g/L proteinase K. DNA loading buffer was added, and the samples were resolved through a 10% TBE-PAGE gel. Gels were quantified using the ImageQuant software and analyzed with the GraphPad Prism software.

**Strains and media.** All bacterial strains were derivatives of *E. coli* K-12 and are described in Supplementary Table 5. The protocol for P1 transduction has been described previously[69]. All P1 transductions were selected on 2% agar plates made with either Luria Broth or 56/2 minimal media (supplemented with 0.2% glucose, 0.001% thiamine and specified amino acids, and containing the appropriate antibiotics[69]). Kanamycin was used at 50 μg/mL, chloramphenicol at 25 μg/mL, ampicillin at 50 μg/mL and tetracycline at 10 μg/mL. All transductants were grown at 37 °C and purified on the same type of media on which they were selected.

**Construction of priA^tm::Asp438Ala (GCT), Thr482Ala (GCT), His483Ala (GCG) on the chromosome.** To place *priA^tm* on the chromosome, first the RsrII-EagI fragment of pSJS1625[47] was exchanged with the corresponding fragment of DNA on pAD036 (plasmid containing *priA^tm::Asp438Ala (GCT), Thr482Ala (GCT), His483Ala (GCG)*). This placed a *cat* gene at the end of *priA* and in-between *priA^tm* and homologous sequences downstream of *priA*. This was done by restricting pAD036 with EagI and RsrII and pSJS1625 with EagI, RsrII, and PstI, combining the DNAs and treating with T4 DNA ligase. This mixture was then used to transform DH5α *E. coli* cells. Chloramphenicol-resistant transformants were selected. One plasmid having the proper configuration was saved and called pSJS1648. pSJS1648 was then restricted with EagI, EcoRV and PstI and used to transform SS9276 (*del(priA)317::kan, dnaC809,820*)[47] with pKD46[70]. *del(priA)317::kan* substitutes a *kan* gene for the last part of *priA* (nt.796 to 2301). Homologous recombination with regions between the EcoRV site in *priA* and the end of the deletion and sequences downstream of *priA* repaired the *priA* deletion mutation with DNA containing *priA^tm* transferred *priA^tm* to the chromosome. This was accomplished through

the selection of chloramphenicol resistant transformants and screening those for kanamycin sensitivity. The presence of a full length *priA* gene and a BstXI restriction pattern consistent with the *priA^tm::cat* allele was verified by PCR. One strain was saved and called SS13594.

**Preparation and analysis of cells for microscopy.** Cells for determining cell size and the levels of SOS expression were prepared, imaged and computer analyzed as previously described[27]. Briefly cells were grown in minimal media to mid-log phase and then 2 μL of that suspension were added to a 2% agarose slab. A coverslip was then applied on top of the cells. The cells were grown at 37 °C for 3 to 4 hours. Images (phase contrast and fluorescence) were taken for at least 9 different fields of view (3 on 3 different days) for each strain. These images were then analyzed using a two-step process. First, the cells were segmented using a program called Omnipose[71]. Then using another program written using Matlab R2022a, the properties of the cells with respect to size and Relative Fluorescence Intensity (RFI) were determined. RFI for each cell was normalized to the average fluorescence intensity of the background. Typically, between 700 and 1500 cells are counted for each strain.

### Reporting summary
Further information on research design is available in the Nature Portfolio Reporting Summary linked to this article.

## Data availability
The cryo-EM map for PriA/PriB/DNA has been deposited in the Electron Microscopy Data Bank (EMDB) with the accession code EMD-28959 and the final model in the Protein Data Bank (PDB) with the identifier 8FAK. Source data for uncropped gel images and gel quantification are provided in the Source Data file. Source data are provided with this paper.

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

## Acknowledgements
The authors thank the Cryo-EM Research Center in the Department of Biochemistry at the University of Wisconsin-Madison for providing electron microscopy facilities and instrumentation. We thank the members of the Keck lab, James Berger, and Ci Ji Lim for critical evaluation of this manuscript. Research reported in this publication was supported by the National Institute of General Medical Sciences of the National Institutes of Health under Award Number T32GM008349 to ATD and HRD and National Institutes of Health grant RO1GM098885 to JLK.

## Author contributions
Conceptualization: A.T.D. and J.L.K. Methodology: A.T.D., P.L.D., S.D.M., S.J.S., T.G., J.L.K. Investigation: A.T.D., P.L.D., S.D.M., K.A.S., K.H.B., H.R.D., J.A.L., S.J.S., T.G. Writing: A.T.D. and J.L.K. Funding Acquisition: S.J.S., T.G., J.L.K.

## Competing interests
The authors declare no competing interests.
