## [Peer Review File · Nature Communications]

Replication fork binding triggers structural changes in the PriA helicase that govern DNA replication restart in E. coliREVIEWER COMMENTS

Reviewer #1 (Remarks to the Author):

Premature replisome dissociation from the replication fork can be lethal. To avoid this, bacterial cells evolved replication restart strategies involving PriA-PriB recruitment. To understand the mechanism, the authors studied the structure of the PriA/PriB in complex with an artificial fork substrate. They used cryo-EM to get a 3.2 Å resolution structure of the complex. The resulting structure showed an extensive interface between PriA each arm in the DNA fork, which confers structural specificity. The binding of the replication fork causes a rearrangement of PriA's structure, creating a pore around the lagging strand DNA and exposing a docking site for PriB. The authors present convincing Biochemical and genetic evidence indicating that PriA's domain rearrangement couples replication fork recognition to PriA/PriB complex formation and that dimer populations observed are artefactual.

I have a few minor comments to:

- 1) I feel Figure 1 should show cryo-EM density and not only the surface representation derived from the atomic model.
- 2) Reference should be added to line 103 page 3 to substantiate the claim.
- 3) Figure S1. Some of the structures appear to contain significant bias in particle orientation. Does this anisotropy result in artificial elongation along the over-represented axis? To alleviate my concerns the authors should display a 90 degree rotated view of each of the three volumes. Also, it is not clear to me why the dimeric structures have angular distribution, FSC plot and isosurface representation shown in smaller insets compared to the monomeric structure. Can the authors elaborate on why they think the FSC curve rises above the 0.143 cutoff in the C2-symmetry-imposed structures?
- 4) In Supplementary Figure 2 DNA interacting residues are shown but their identify is not defined and the panels are too small to be informative. Also, I believe in this figure the authors miss out on the opportunity of showing the quality of the density in critical protein-DNA interacting regions. Only one instance is shown in Supplementary Figure 1C, but it is not clear from the panel alone what region this density corresponds to.
- 5) PriA/PriB-mediated DnaB loading assays appear to be very weak signals. The authors mention the low efficiency is a result of "complexity of the reconstituted system" could the authors elaborate on this? Could the problem be with challenges in staging the reaction? Could the length of the fork arms make a difference?
- 6) The authors claim in the abstract that bacterial replisomes 'often dissociate from replication forks'. They later claim that 'abandoned DNA replication forks are relatively rare'. This seems inconsistent.

Reviewer #2 (Remarks to the Author):

This study reports a 3.2 Å structure of an *E. coli* PriA-PriB-replication fork complex. Replication fork binding induces a conformational change in PriA that creates a pore enclosing the lagging strand. A PriB binding domain is also repositioned which enables PriB to bind ssDNA extruding from the lagging strand pore. The authors investigate several binding mutants to probe the importance of these interactions and assess their impact on complex formation, stimulation of PriA helicase activity, replicative helicase reloading, and *in vivo* DNA replication defects. A "switch-like mechanism model for replication restart initiation" is proposed where replication fork binding remodels PriA which causes PriB recruitment and PriB-mediated replication restart.

This paper is very well written. The introduction is concise and most of the arguments in the results and discussion are easy to follow. As a non-structural biologist I found the complex descriptions and conclusions in the first results sections to be particularly well-explained. The paragraph beginning on line 245 was more confusing because it was unclear which dimer class was being investigated.

The central conclusions are significant to the DNA replication field and are well supported by the results. This paper is a beautiful example of structural insight leading to a testable and useful model. Characterising replication restart at the mechanistic level is important for understanding genome stability and will therefore be of interest to researchers in a variety of fields. All the genetics and biochemistry experiments have been carried out to a high standard using appropriate methods. Figures are clear and enough methodological detail is provided to enable this work to be repeated. Conclusions are justified well throughout. As a non-specialist I am unable to properly judge the cryo-EM methodology and analysis. However, I was impressed by the extensive checking of both minor dimer cryo-EM classes to justify their dismissal as artifacts.

I genuinely enjoyed reading this paper and I commend the authors on producing a high quality and interesting story.

Specific comments

60: Citing a PriA review would be helpful.

246: If the possibility of PriA binding to both PriB protomers is being tested then should this be Figure S3B rather than S3A?

265: The mobility shift is similar in size but is not as efficient because the lower band for PriA only is much stronger. The mutants therefore retain DNA binding activity but at a lower level. Please add a comment to reflect this weaker DNA binding.

371: Would appreciate some speculation as to how modest replication restart activities are maintained in *priAtm/priC*-null cells.

464: Could similar strategies be used in archaeal systems?

499: Should be PriB conservation.

Reviewer #3 (Remarks to the Author):

In their work entitled « Replication fork binding triggers structural changes in the PriA helicase that govern DNA replication restart in *E. coli* », Duckworth et al report the cryogenic electron microscopy structure of a PriA/PriB/replication fork complex. PriA is a DNA helicase, which mediates the essential process of restarting prematurely terminated DNA replication reactions in bacteria, including in *E. coli*. The structure reveals extensive interactions between PriA interactions and each arm of the branched DNA. These interactions reshape the PriA protein to create a pore encircling single-stranded lagging-strand DNA and promotes a conformational change that favors binding to PriB. The structural study is supported by biochemical and genetic studies, and aims to gain insight into the mechanism for replication restart, a fundamental biological question in all three domains of life.

This new cryo-EM structure, as well as the biochemical and genetic studies are interesting per se. Nevertheless, the way in which the results are presented needs to be substantially improved. In its current state, the structural analysis is insufficient and does not allow the reader to fully appreciate the contribution of this new structure. In addition, several claims regarding the recruitment of PriB by PriA and its importance on the recruitment of DnaB needs to be clarified.

My comments are listed below:

1) PriA/PriB/replication fork structure:

a. Figure 1: PanelA: The surface representation of PriA is particularly misleading. The cryo-EM model (map and cartoon representation) should be represented in at least one of the main figure. The density surrounding the DNA fork should also be represented in order for the reader to appreciate the quality of the model in the different parts of the DNA fork. The polarity of the DNA strands should be indicated on the figure. The parental DNA, the lagging and leading strand may be highlighted by using different color codes. Panel 1B: The authors claim that “replication fork binding triggered a dramatic rearrangement of the PriA structure ...”. It is very hard to catch the structural rearrangements that occur between the crystal apo structure and the DNA-bound cryo-EM structure in the current figure.

Adding arrows to highlight the domain motion of the PriA-CRR domain, and possibly a sketch, may help. Panel 1C: The ssDNA binding pore is hardly visible in this orientation. Maybe a cutaway view of the complex, highlighting the structural rearrangements would help.

b. Cryo-EM data: The quality of the cryo-EM data and the reported average resolution (3.2Å) are correct. However, the cryo-EM structure determination work-flow is not sufficiently detailed. Key missing informations are: the local resolution maps and 3D-FSCs. In the validation report (page 24-§9.1), the map-model overlay shows that some regions of the model are not covered by the map. The local resolution maps would enable the reader to appreciate, which regions of the model are ordered or more flexible.

c. Cryo-EM sample preparation: Neither ATP, nor non-hydrolysable ATP was added to the complex, which was used for determining the cryo-EM structure of the PriA/PriB/replication fork structure. However, under physiological conditions, the ATPase domain must exist in complex with ATP or hydrolyzed ATP. Would the structure of the complex presented in this manuscript be altered by ATP binding? The authors previously reported a study showing that " An aromatic-rich loop couples DNA binding and ATP hydrolysis in the PriA DNA helicase ". What is the conformation of this loop in the cryo-EM structure of the PriA/PriB/replication fork structure?

2) PriA/PriB complex formation:

a. Figure 2: Panel A should include the distances measured for the interfacial interactions. The map surrounding the key residues should also be represented in the main figure or in supplementary material. Surface representations in panel 2B are useful but the authors should also show an overall structure of the interface between PriA-CRR and PriB so that the readers can visualize this interfacial region in the context of the entire complex.

b. Cross-linking experiments: One may expect a stronger cross-linking of PriB to DNA in presence of PriA. However, the amount of cross-linked PriB (highlighted by the intensity of the bands) seems higher in absence than in presence of PriA. This result is rather unexpected and may not be solely explained by the Bpa preference for cross-linking thymine nucleobases. Therefore, this results should be discussed further.

3) PriA variants: The site-directed mutagenesis experiments and biochemical assays are sound and well executed. For clarity, the authors should homogenize the nomenclatures used in the main text and in the main figures. Several sentences should be rephrased: e.g. lines 278-279 (page 7) and 305-209 (page 8). In figure 4: the usage of the three-letters code or the one-letter code for designing the mutations should be homogenized. Same for triple variant, or (D438A, T482A, H483A) nomenclatures used in the same figure.

4) PriA/PriB-mediated DnaB reloading: Only 10% of the substrate is unwound at the highest PriB concentration tested. The authors hypothesize that this is due to the complexity of the reconstituted system. This may be true. Nevertheless this experiment should be interpreted more carefully and their statement that: "the PriA-CRR/PriB interface appears critical for loading DnaB onto DNA replication forks (lines 352-353)" should be tuned down.

Point-by-point responses to reviewer comments

Reviewer #1 (Remarks to the Author):

Premature replisome dissociation from the replication fork can be lethal. To avoid this, bacterial cells evolved replication restart strategies involving PriA-PriB recruitment. To understand the mechanism, the authors studied the structure of the PriA/PriB in complex with an artificial fork substrate. They used cryo-EM to get a 3.2 Å resolution structure of the complex. The resulting structure showed an extensive interface between PriA each arm in the DNA fork, which confers structural specificity. The binding of the replication fork causes a rearrangement of PriA's structure, creating a pore around the lagging strand DNA and exposing a docking site for PriB. The authors present convincing Biochemical and genetic evidence indicating that PriA's domain rearrangement couples replication fork recognition to PriA/PriB complex formation and that dimer populations observed are artefactual.

I have a few minor comments to:

1) I feel Figure 1 should show cryo-EM density and not only the surface representation derived from the atomic model.

We agree and have moved the cryo-EM density map from Figure S1B (in the original submission) to Figure 1 in the revised manuscript.

2) Reference should be added to line 103 page 3 to substantiate the claim.

We have added three references with experiments demonstrating that PriA binds DNA forks in a structure specific manner and recruits PriB: Liu et al. (1996) *J Biol Chem* **271**, 15656-15661, Windgassen et al. (2018) *Proc Natl Acad Sci* **115**, 9075-9084, and Lopper et al. (2007) *Mol Cell* **26**, 781-793.

3) Figure S1. Some of the structures appear to contain significant bias in particle orientation. Does this anisotropy result in artificial elongation along the over-represented axis? To alleviate my concerns the authors should display a 90 degree rotated view of each of the three volumes. Also, it is not clear to me why the dimeric structures have angular distribution, FSC plot and isosurface representation shown in smaller insets compared to the monomeric structure. Can the authors elaborate on why they think the FSC curve rises above the 0.143 cutoff in the C2-symmetry-imposed structures?

Indeed, the dimer structures do have significant orientation bias and are generally quite low quality. The original methods text attempted to highlight this by stating “Both dimer forms exhibited strong preferential orientation and despite relatively high global resolutions as measured by the Fourier shell correlation (FSC) and 0.143 cut-off (3.8 Å and 3.9 Å) the overall quality of the maps is quite poor with features at resolutions higher than ~7 Å resolution only visible in one specific orientation. Given the poor quality of the maps and the fact that at the obtained resolutions they both appear to be two copies of the monomer form, further investigation was restricted to the monomer form.” This combined with the *cis*TEM way of measuring FSCs led to the large high-resolution fluctuations in the FSC. We have added 3D FSCs and local resolution renderings which show the maps from different directions to Figure S1 and made all three panels equal in size which we hope will make the orientation problems clearer.

4) In Supplementary Figure 2 DNA interacting residues are shown but their identify is not defined and the panels are too small to be informative.

We agree that this will help to define the protein/DNA interface for readers and have added a new panel to Figure S2 (S2B) that shows a schematic of all PriA and PriB residue/DNA

interactions, including the identities of each DNA-interacting residue and the specific sites of interaction for each nucleotide (base, sugar, or phosphate).

Also, I believe in this figure the authors miss out on the opportunity of showing the quality of the density in critical protein-DNA interacting regions. Only one instance is shown in Supplementary Figure 1C, but it is not clear from the panel alone what region this density corresponds to. We have expanded Figure S1 to include additional images that are labeled to show the model and associated EM density of protein-DNA interaction regions.

5) PriA/PriB-mediated DnaB loading assays appear to be very weak signals. The authors mention the low efficiency is a result of “complexity of the reconstituted system” could the authors elaborate on this? Could the problem be with challenges in staging the reaction? Could the length of the fork arms make a difference?

The complexity of the DnaB-loading assay likely stems from the fact that there are six proteins required for activity (PriA, PriB, DnaT, DnaB, DnaC, and SSB). Each protein has an optimal concentration and the concentration balance between each of the proteins needs to be finely tuned relative to the others (Manhart and McHenry (2013) *J Biol Chem* **288**, 3989-3999). Having too much or too little of a given protein affects the overall efficiency of DnaB loading. The length of the lagging strand may make a difference as well – in this paper, we have used the substrate that has been used previously (in Heller and Marians (2005) *Mol Cell* **17**, 733-743 and Lopper et al. (2007) *Mol Cell* **26**, 781-793) and that was used for the other assays in our manuscript to allow for direct comparisons.

DnaB loading assays have been published three separate times (Heller and Marians (2005) *Mol Cell* **17**, 733-743; Lopper et al. (2007) *Mol Cell* **26**, 781-793; Manhart and McHenry (2013) *J Biol Chem* **288**, 3989-3999) and all have reported ~10-20% maximal efficiencies. Despite the limited overall signal that comes from DnaB-mediated DNA unwinding in the assay, the difference between wildtype and triple variant PriA activities are apparent and statistically significant in Figure 5. Importantly, the findings also correlate well with the other *in vitro* and *in vivo* data presented in the paper. Nonetheless, the comment from the reviewer and a similar comment by reviewer #3 have led us to reduce the strength of our interpretation of the DnaB loading results in our revised manuscript and we thank both reviewers for pointing out the shortcomings of this assay.

6) The authors claim in the abstract that bacterial replisomes ‘often dissociate from replication forks’. They later claim that ‘abandoned DNA replication forks are relatively rare’. This seems inconsistent.

We thank the reviewer for pointing out this paradox. The first line of the abstract is referring to the fact that replication forks are thought to be abandoned roughly once per replication cycle. We later say the forks are “relatively rare” because the relative concentration of abandoned forks in the cell is likely to be at most one per cell. We have reworded our latter statement to avoid this inconsistency by changing the later sentence to “Given that abandoned DNA replication forks are present at a relatively low concentration in cells (occurring a small number of times per cell cycle²⁻⁴), this form may be an artifact of sample preparation”.

Reviewer #2 (Remarks to the Author):

This study reports a 3.2 Å structure of an *E. coli* PriA-PriB-replication fork complex. Replication fork binding induces a conformational change in PriA that creates a pore enclosing the lagging strand. A PriB binding domain is also repositioned which enables PriB to bind ssDNA extruding from the lagging strand pore. The authors investigate several binding mutants to probe the importance of these interactions and assess their impact on complex formation, stimulation of PriA helicase activity, replicative helicase reloading, and *in vivo* DNA

replication defects. A “switch-like mechanism model for replication restart initiation” is proposed where replication fork binding remodels PriA which causes PriB recruitment and PriB-mediated replication restart.

This paper is very well written. The introduction is concise and most of the arguments in the results and discussion are easy to follow. As a non-structural biologist I found the complex descriptions and conclusions in the first results sections to be particularly well-explained. The paragraph beginning on line 245 was more confusing because it was unclear which dimer class was being investigated.

The central conclusions are significant to the DNA replication field and are well supported by the results. This paper is a beautiful example of structural insight leading to a testable and useful model. Characterising replication restart at the mechanistic level is important for understanding genome stability and will therefore be of interest to researchers in a variety of fields. All the genetics and biochemistry experiments have been carried out to a high standard using appropriate methods. Figures are clear and enough methodological detail is provided to enable this work to be repeated. Conclusions are justified well throughout. As a non-specialist I am unable to properly judge the cryo-EM methodology and analysis. However, I was impressed by the extensive checking of both minor dimer cryo-EM classes to justify their dismissal as artifacts.

I genuinely enjoyed reading this paper and I commend the authors on producing a high quality and interesting story.

We thank the reviewer for their kind words and enthusiastic review.

Specific comments

60: Citing a PriA review would be helpful.

We have added citations to two reviews: A PriA-centric review (Gabbai and Marians (2010) *DNA Repair* **9**, 202-209) and a DNA replication restart review that focuses on molecular mechanisms (Windgassen et al. (2018) *Nucleic Acids Res* **46**, 504-519).

246: If the possibility of PriA binding to both PriB protomers is being tested then should this be Figure S3B rather than S3A?

To alleviate the confusion between the dimer forms mentioned in the paper, we have renamed the two dimer classes as Dimer 1 and Dimer 2 and updated all references to these structures. Dimer 1 contains two PriA molecules bound to a PriB homodimer via the PriA-CRR. Dimer 2 contains the PriA-CTD/PriB interface.

The experiment referenced in line 246 is testing if PriA needs to bind to both sides of a PriB homodimer for activity. Thus, this is testing the Dimer 1 structure in Fig. S3A.

265: The mobility shift is similar in size but is not as efficient because the lower band for PriA only is much stronger. The mutants therefore retain DNA binding activity but at a lower level. Please add a comment to reflect this weaker DNA binding.

This discrepancy is likely due to experimental error (small inaccuracies in diluting PriA to 2 nM or the fork to 1 nM) rather than abated binding activity. After quantifying the 0 nM PriB lanes from all replicates, we observe that WT shifts $91.9 \pm 4.1\%$ of the labeled fork, D438A shifts $95.0 \pm 1.9\%$, T482A shifts $91.3 \pm 6.5\%$, H483A shifts $92.2 \pm 3.7\%$, and the triple variant shifts $95.8 \pm 3.1\%$. We have added a note with these data to the manuscript.

As a visual confirmation of this variation, one can compare the WT lanes in Fig. 4A and Fig S3E. Slight differences in the amount of free DNA are apparent between the two individual lanes. These differences are represented in the percent error from multiple experiments.

371: Would appreciate some speculation as to how modest replication restart activities are maintained in priAtm/priC-null cells.

We also find this observation interesting. We hypothesize that the modest replication restart activity may be due to residual binding of PriB to the PriA-CRR triple variant *in vivo* that is not detected *in vitro*. It is possible, for example, that SSB bound to the lagging strand in cells recruits PriB to the replication fork through the known SSB-PriB interaction (Low, Shlomai & Kornberg (1982) *J Biol Chem* **257**, 6242). Since the lagging strand in cells can be >1000 bases long with many bound SSBs, the local concentration of PriB associated with SSB at the replication fork may be higher in cells than in the *in vitro* assays used in our manuscript. Such a phenomenon could bolster a weak interaction between PriB and the PriA-CRR triple variant, allowing modest PriA-PriB pathway activity in *priAtm* cells. We have noted this possibility in the revised manuscript.

464: Could similar strategies be used in archaeal systems?

Yes, we believe that this general mechanism could be conserved in archaea and in eukaryotes. While there are no PriA homologs in archaea or eukaryotes, fork recognition and repair are essential processes in all domains of life. Thus, the PriA-CRR regulatory switch we discuss in the manuscript could be a generalized mechanism employed by DNA repair proteins in the other domains to recognize the “correct” repair substrate. We have updated this line to include this view.

499: Should be PriB conservation.

It is correct as PriA conservation. As noted, PriB is not well-conserved. Thus, when calculating PriA conservation scores, we filtered the PriA sequence list to include only those from bacteria with an annotated PriB gene. If we did not do this, the scores might include PriA sequences that do not have a PriB-binding interface and thus lower the conservation scores of residues important for PriB-binding.

Reviewer #3 (Remarks to the Author):

In their work entitled « Replication fork binding triggers structural changes in the PriA helicase that govern DNA replication restart in *E. coli* », Duckworth et al report the cryogenic electron microscopy structure of a PriA/PriB/replication fork complex. PriA is a DNA helicase, which mediates the essential process of restarting prematurely terminated DNA replication reactions in bacteria, including in *E. coli*. The structure reveals extensive interactions between PriA interactions and each arm of the branched DNA. These interactions reshape the PriA protein to create a pore encircling single-stranded lagging-strand DNA and promotes a conformational change that favors binding to PriB. The structural study is supported by biochemical and genetic studies, and aims to gain insight into the mechanism for replication restart, a fundamental biological question in all three domains of life.

This new cryo-EM structure, as well as the biochemical and genetic studies are interesting per se. Nevertheless, the way in which the results are presented needs to be substantially improved. In its current state, the structural analysis is insufficient and does not allow the reader to fully appreciate the contribution of this new structure. In addition, several claims regarding the recruitment of PriB by PriA and its importance on the recruitment of DnaB needs to be clarified.

My comments are listed below:

1) PriA/PriB/replication fork structure:

a. Figure 1:

i. PanelA:

1. The surface representation of PriA is particularly misleading. The cryo-EM model (map and cartoon representation) should be represented in at least one of the main figure.

We agree and have moved the cryo-EM density map from Figure S1B (in the original submission) to Figure 1 in the revised manuscript.

2. The density surrounding the DNA fork should also be represented in order for the reader to appreciate the quality of the model in the different parts of the DNA fork.

The change noted above has moved the EM density for the entire fork, including areas that are lost at higher sharpening levels due to mobility, to Figure 1. The figure legend text states “A 3.2 Å resolution cryo-EM density map (surface, colored to reflect modeled PriA and PriB domain positions within the density) overlaid onto a map low-pass filtered to ~5 Å resolution map (mesh). The filtered map highlights the extended DNA regions which are not visible at higher sharpening due to their inherent flexibility and thus lower resolution.” We have also added two new panels to the supplement (Figure S1D) that show EM density for PriA/DNA interfaces.

3. The polarity of the DNA strands should be indicated on the figure. The parental DNA, the lagging and leading strand may be highlighted by using different color codes.

We thank the reviewer for point out this omission. We have labeled the polarity of the DNA strands in the model in Figure 1. Given the large number of colors that are already used in the structure figure, we have opted to keep the DNA a single color and have instead created a new supplementary figure that clearly depicts all interactions between the proteins and each strand of the DNA (Figure S2B).

4. Panel 1B: The authors claim that “replication fork binding triggered a dramatic rearrangement of the PriA structure ...”. It is very hard to catch the structural rearrangements that occur between the crystal apo structure and the DNA-bound cryo-EM structure in the current figure. Adding arrows to highlight the domain motion of the PriA-CRR domain, and possibly a sketch, may help.
This a great suggestion - we have added an arrow to this figure to highlight the domain motion.

- ii. Panel 1C: The ssDNA binding pore is hardly visible in this orientation. Maybe a cutaway view of the complex, highlighting the structural rearrangements would help.

Figure S2A features a cutaway view of the pore.

- b. Cryo-EM data: The quality of the cryo-EM data and the reported average resolution (3.2Å) are correct. However, the cryo-EM structure determination work-flow is not sufficiently detailed. Key missing informations are: the local resolution maps and 3D-FSCs. In the validation report (page 24- §9.1), the map-model overlay shows that some regions of the model are not covered by the map. The local resolution maps would enable the reader to appreciate, which regions of the model are ordered or more flexible.

We agree and have added 3D FSCs and local resolution maps to the Figure S1A to better allow readers to judge the map quality.

- c. Cryo-EM sample preparation: Neither ATP, nor non-hydrolysable ATP was added to the complex, which was used for determining the cryo-EM structure of the PriA/PriB/replication fork structure. However, under physiological conditions, the

ATPase domain must exist in complex with ATP or hydrolyzed ATP. Would the structure of the complex presented in this manuscript be altered by ATP binding? The authors previously reported a study showing that " An aromatic-rich loop couples DNA binding and ATP hydrolysis in the PriA DNA helicase ". What is the conformation of this loop in the cryo-EM structure of the PriA/PriB/replication fork structure?

ATP was not added to the samples to avoid unwinding of the DNA substrate. PriA preferentially unwinds the lagging strand and can also unwind parental DNA with very low efficiency, thus we would likely not be able to trap a PriA/PriB/replication fork DNA complex in the presence of ATP.

The conformation of the aromatic-rich loop in our structure is similar to all published structures and we have now noted this in the revised manuscript. It would be an interesting experiment to see if adding a non-hydrolyzable ATP analog would alter the structure of PriA, especially the aromatic-rich loop. Based on examples of other superfamily 2 helicases, we predict that the addition of an ATP analog could lead to small structural changes in the helicase domains.

2) PriA/PriB complex formation:

a. Figure 2:

- i. Panel A should include the distances measured for the interfacial interactions.

We have added bond distances to this panel.

- ii. The map surrounding the key residues should also be represented in the main figure or in supplementary material.

We have added a panel to Figure S1 (S1C) which depicts the model and density map of the PriA/PriB interface.

- iii. Surface representations in panel 2B are useful but the authors should also show an overall structure of the interface between PriA-CRR and PriB so that the readers can visualize this interfacial region in the context of the entire complex.

We agree and have modified Figure 2 to include a global image of the PriA/PriB interface.

- b. Cross-linking experiments: One may expect a stronger cross-linking of PriB to DNA in presence of PriA. However, the amount of cross-linked PriB (highlighted by the intensity of the bands) seems higher in absence than in presence of PriA. This result is rather unexpected and may not be solely explained by the Bpa preference for cross-linking thymine nucleobases. Therefore, this results should be discussed further.

It is hard to directly determine crosslinking efficiency from the primer extension assay (Figure 3) due to the crosslinking products being spread out across the 60nt region of ssDNA. However, the amount of un-crosslinked product where the polymerase was not blocked (full-length product) looks to be similar in the presence or absence of PriA. Possibly since PriA is blocking the site of most intense crosslinking (the fork junction), we lose some signal to the crosslinks being spread out. Moreover, in Figure S4, the amount of crosslinked products in Panels B-D is similar in the +/- PriA lanes. Overall, we do not think PriA has any significant effect on crosslinking efficiency. It does, however, affect the location of PriB crosslinking, as discussed in the manuscript.

3) PriA variants: The site-directed mutagenesis experiments and biochemical assays are sound and well executed. For clarity, the authors should homogenize the nomenclatures used in the main text and in the main figures.

a. Several sentences should be rephrased: e.g. lines 278-279 (page 7) and 305-209 (page 8).

These sentences have been rephrased. For clarity, the N-term Arg44Ala LD and C-term Arg44Ala LD are now collectively referred to as “the single Arg44Ala LD PriB variants”.

b. In figure 4: the usage of the three-letters code or the one-letter code for designing the mutations should be homogenized. Same for triple variant, or (D438A, T482A, H483A) nomenclatures used in the same figure.

Figures 4, 5, S3E-L, and S6 have been updated to only include three-letter amino acid codes. All references to amino acids in the text or figures now use the three-letter codes only.

4) PriA/PriB-mediated DnaB reloading: Only 10% of the substrate is unwound at the highest PriB concentration tested. The authors hypothesize that this is due to the complexity of the reconstituted system. This may be true. Nevertheless this experiment should be interpreted more carefully and their statement that: “the PriA-CRR/PriB interface appears critical for loading DnaB onto DNA replication forks (lines 352-353)” should be tuned down.

We thank the reviewer for this insight. Based on this comment and a similar comment from reviewer #1, we have modified our interpretation of the DnaB reloading assay results. The passage now reads: “As with our prior PriB binding and helicase results, no DnaB loading activity was observed for the PriA-CRR triple variant at any PriB concentration tested. Thus, the PriA-CRR/PriB interface appears important for loading DnaB onto DNA replication forks.”

REVIEWERS' COMMENTS

Reviewer #1 (Remarks to the Author):

The authors addressed all of my concern with their improved manuscript. Congratulations on some excellent work.

Reviewer #2 (Remarks to the Author):

The authors have addressed all my points with their explanations and additions to the text. They have also made some global nomenclature changes that will help readers of this paper. Finally, as far as I can judge, the authors have made considerable and helpful changes to the manuscript to alleviate concerns and address comments made by the other reviewers.

I am happy for this revised manuscript to be published.

Reviewer #3 (Remarks to the Author):

In their revised manuscript, the authors have successfully answered to all my concerns. In particular, the important structures presented in this interesting work are now nicely illustrated by the figures in their revised forms.

Minor comment: There are two shapes hiding the side chains of the two histidines in Figure 2, panel A.

RESPONSE TO REVIEWERS' COMMENTS:

Reviewer #1 (Remarks to the Author):

The authors addressed all of my concern with their improved manuscript. Congratulations on some excellent work.

We thank the reviewer for their very positive review.

Reviewer #2 (Remarks to the Author):

The authors have addressed all my points with their explanations and additions to the text. They have also made some global nomenclature changes that will help readers of this paper. Finally, as far as I can judge, the authors have made considerable and helpful changes to the manuscript to alleviate concerns and address comments made by the other reviewers.

I am happy for this revised manuscript to be published.

We thank the reviewer for their very positive review.

Reviewer #3 (Remarks to the Author):

In their revised manuscript, the authors have successfully answered to all my concerns. In particular, the important structures presented in this interesting work are now nicely illustrated by the figures in their revised forms.

We thank the reviewer for their very positive review.

Minor comment: There are two shapes hiding the side chains of the two histidines in Figure 2, panel A.

We have now corrected this in the revised figure 2.